# Contemporary Treatment of Crime Victims/Survivors: Barriers Faced by Minority Groups in Accessing and Utilizing Domestic Abuse Services [note 1]

**DOI:** 10.3390/bs15020103

**Published:** 2025-01-21

**Authors:** Terri Cole, Orlanda Harvey, Jane C. Healy, Chloe Smith

**Affiliations:** 1Department of Psychology, Faculty of Science and Technology, Talbot Campus, Bournemouth University, Poole BH12 5BB, UK; chloesmith7366@gmail.com; 2Department of Social Sciences and Social Work, Faculty of Health and Social Science, Lansdowne Campus, Bournemouth University, Poole BH8 8GP, UK; harveyo@bournemouth.ac.uk (O.H.); jhealy@bournemouth.ac.uk (J.C.H.)

**Keywords:** victim, survivor, support, domestic abuse, domestic violence, intimate partner violence, LGBTQIA+, black and ethnic minority, disabled people, minority groups

## Abstract

This research explored the experiences of LGBTQIA+, black and ethnic minority (BME), and disabled victims of domestic abuse due to the frequency of abuse in these populations and bespoke needs they may have. Data were collected via an online survey (n = 317), a focus group with professionals (n = 2), and interviews with victims/survivors of domestic abuse (n = 2). Many victims/survivors articulated difficulties in accessing support for many reasons, including individual and structural barriers such as embarrassment, stigma, shame, fear and not being aware of what support is available. Whilst good practice was reported, examples of secondary victimization towards victims/survivors by individuals, professionals and organizations were recounted. Many barriers were identified; for example, there was inappropriate provision in refuges or shelters for LGBTQIA+ groups or disabled people. Disabled victims experienced additional barriers if their abuser was also their carer. BME groups may have additional language difficulties as well as cultural stigma and pressure to stay with their abuser. Recommendations for practice include the need for enhanced multi-agency training and recognition of abuse; crime against victims/survivors being supported by someone with the same cultural background; easier access to interpreters; and more appropriate refuge or alternative housing options.

## 1. Introduction

### 1.1. Definitions and Prevalence

Gender-based violence incorporates a spectrum of sexual, violent and abusive behaviors and can occur in relationships, friendships, or between complete strangers. Domestic abuse (DA) is defined as an incident or pattern of incidents of controlling, coercive, threatening, degrading and/or violent behavior, usually perpetrated by a partner or ex-partner, but can also be familial, such as by a family member or carer ([14]). Thus, intimate partner violence and abuse (IPVA) is part of DA, but only in circumstances where individuals are partners or former partners, such as those who are or have been in an intimate relationship with each other ([15]). It is estimated 2.1 million people (1.4 million women and 751,000 men) in England and Wales experienced domestic abuse in the year ending March 2023 ([39]), the vast majority of whom were victims/survivors of IPVA, and the prevalence data from 161 countries found nearly one in three women have been subjected to either physical and/or sexual intimate partner violence or non-partner sexual violence ([51]).

DA/IPVA occurs in all society; however, emergent research suggests increased prevalence within certain minority communities such as the following: LGBTQIA+ ([36]); BME ([30]); and disabled people ([35]). In response, bespoke services are being commissioned to explore particular barriers and issues such groups may face in order to address specific needs. This research was commissioned to consider the potential utility of bespoke victim/survivor services for such groups currently being provided by two charities in Southampton, UK (Yellow Door and Stop Domestic Abuse). The research focus was DA and not restricted to IPVA only. The authors acknowledge discussions regarding terminology of abuse ([43]); therefore, throughout this paper will use the all-encompassing term victims/survivors.

#### 1.1.1. LGBTQIA+ Victims/Survivors

Prevalence statistics do not always include sexuality or gender identity; however, a national survey of the Lesbian Gay Bisexual Transgender Queer Intersex Asexual (LGBTQIA)+ community in the UK found one in four lesbian and bisexual women (25%) have experienced DA ([23]). In relation to IPVA, research found nearly 12% of gay men reported experiencing physical violence ([49]); another study reported 86% of young gay men had been subject to psychological aggression, including 67% experiencing physical assault and 64% experiencing sexual coercion ([34]). Disparity between reporting levels may be due to different methods and samples; however, the overall prevalence is alarming. Others have found high rates of DA reported by bisexual people; a lifetime prevalence for rape, physical violence, and stalking was reported by 61% of bisexual women and 27% of bisexual men ([11]), and [41] ([41]) found transgender individuals were twice as likely to experience DA than their cisgendered counterparts. Another study found more than two-thirds of female-to-male individuals had experienced DA ([37]).

#### 1.1.2. BME Victims/Survivors

The Crime Survey for England and Wales ([39]) reports a higher percentage of victims/survivors of DA were black and minority ethnic (BME). Others have found ([31]) BME groups made up nearly 13% of victims/survivors of psychological abuse and over 4% physical abuse. [3] ([3]) found women from communities in which family abuse is normalized were less likely to seek help in order to protect the family’s reputation. In Pakistan, one study found only 14% of 7897 women who experienced physical abuse reported this to police, with most not reporting to anyone outside their family ([5]). Moreover, [29] ([29]) noted that the barriers to escaping such relationships are exacerbated for women from minority backgrounds. Similarly, in a US study, many migrant women choose not to leave relationships due to cultural and religious ties ([4]). Resultingly, UK organizations supporting victims/survivors are seeking partnerships with specialist services that provide culturally sensitive support for BME communities ([40]).

#### 1.1.3. Disabled Victims/Survivors

The term disabled person is defined as someone with “a physical or mental impairment which has a substantial and long-term adverse effect on their ability to carry out normal day-to-day activities” ([17]). The term disability covers a variety of conditions, impairments and disorders which may enhance the risks of experiencing DA. For example, [18] ([18]) found that 38% of people with physical disabilities experienced physical abuse and 23% economic abuse. Disabled women are 30% more likely to experience DA compared to non-disabled women ([16]). Nineteen percent of individuals with learning disabilities had experienced DA in the year prior to data collection ([38]) and adults with Autistic Spectrum Condition (ASC) were nearly twice as likely to experience DA than those without such diagnoses ([22]). One sample of 79 people with schizophrenia found 73% had experienced DA ([2]). Additionally, over 70% of hearing-impaired women had experienced psychological abuse ([32]) and 1 in 12 visually impaired individuals had experienced DA ([12]). As such, there is significant evidence that disabled women are at increased risk of DA ([8]; [25]; [28]; [42]; [50]).

In summary, accurate levels of prevalence are limited due to a lack of reporting; however, findings suggest these minoritized communities experience higher rates of DA.

### 1.2. Accessing Support: Barriers for Different Communities

Although support is provided through statutory bodies, the charity sector has played a critical role in responding to DA in recent years, working with both victims/survivors and perpetrators ([1]; [40]). Yet, despite such experience and an awareness of limitations (e.g., [20]), recent research highlights many barriers are still faced by victims/survivors in accessing support ([26]), including shame, difficulty recognizing the abuse and ongoing behavior by the abuser ([13]; [26]). These barriers may be exacerbated in specific communities. The problematic nature of gendered stereotypes in encounters of DA may lead to enhanced difficulties in accessing help for the different communities. In relation to the LGBTQIA+ community, some lesbian victims/survivors report being arrested rather than supported, because they were perceived as ‘butch’ or more ‘masculine’ in their relationship ([24]). Physical abuse towards gay men may be ignored as features such as ‘rough sex’ may be perceived as ‘normal’ in such relationships ([36]). Others may not want to contribute to negative perceptions of the LGBTQIA+ community by reporting ([46]). In relation to BME individuals, [44] ([44]) found these victims/survivors take 1.5 times longer to seek help than those who identify as white, and difficulties may be particularly acute for migrant women ([21]). Reasons that are common amongst many victims, such as the potential fear about leaving their relationship ([29]) and that their children will be removed ([6]; [7]), are exacerbated by cultural sensitivities that encourage victims/survivors to keep things ‘within the family’ ([45]). For disabled victims/survivors who are reliant on their abuser for their care, DA can be even more difficult to escape ([47]). Structural factors such as a lack of accessible refuge provision can further prohibit their leaving ([50]).

### 1.3. Purpose of the Current Research

There is very limited previous research investigating the bespoke needs of such communities when it comes to treatment, care and support for DA. As such the focus of this research was to establish how specialist services can be tailored to encourage help-seeking and identify the bespoke needs these groups may have.

The following three research questions were explored:What do people think is good about the current DA support services, particularly in relation to fostering inclusion for LBGTQIA+, BME or disabled victims/survivors?How do people think DA support services for victims/survivors could be improved, particularly in relation to challenges for LBGTQIA+, BME or disabled people?Do LBGTQIA+, BME or disabled people have specific challenges and support needs?

## 2. Materials and Methods

Three simultaneous studies were conducted to the explore experiences and needs of survivors/victims, drawing on phenomenology as a methodological framework and that focused on understanding the victims/survivors’ lived experiences.

### 2.1. Surveys—General Public

An online survey was sent to adults, focusing on those living in Southampton (UK). It was disseminated via social media, and via a total of 88 charities and community groups specializing in support for LGBTQIA+, BME and disabled people. It was designed using JISC Online Surveys and was ‘live’ from September–December 2023. The survey included a mix of tick box and free text answers; for example, a tick box for the following question was included: ‘Was there anything which stopped you from getting help or using support services for domestic abuse?’ (yes—please describe; no; do not know; prefer not to answer). A free text example included in the survey is as follows: ‘Was there any support you would have liked but did not receive?’. The survey collated experiences of DA, asking views on support needs for victims/survivors, with questions regarding barriers to accessing support. It was completed by those who had and had not experienced DA and those who had and had not utilized support services. The survey collected demographic information in relation to gender identity, sexual orientation, age, disability, ethnicity and sexual orientation. This information was used both to classify survey responses, and ensure responses were representative of a variety of individuals.

### 2.2. Focus Group—Practitioners

Despite contacting 30 organizations representing the minority groups and providers of DA support, only two participants attended an arranged session. They were asked questions about their organizations and roles, what works well with specific communities, what needs they may have, what could be improved and any barriers to obtaining support.

### 2.3. Interviews—Victims/Survivors

The research design included interviewing people with lived experiences from each of the three groups (LGBTQIA+, BME, and disabled people) who had accessed DA services; however, only two peopled agreed to be interviewed. Interviews were semi-structured, asking questions regarding the support received, if and how the support helped them, what could have been improved, barriers to getting support and specific needs they had. They were conducted via Zoom and audio recorded with permission of the interviewee. Support was provided directly after the interview and information regarding additional support provided.

### 2.4. Ethics

Full ethical approval was obtained in line with the Bournemouth University ethical codes and the UKRI’s research ethics framework. Representatives from commissioning organizations reviewed draft research designs, all questions were optional for participants, and pseudonyms were used throughout to maintain anonymity.

### 2.5. Analysis

The fieldwork consisted of quantitative data, from ‘tick box’ survey responses and qualitative data from free-text survey answers, plus the focus group and interview data. Qualitative data are rich, textual data that are drawn from interviews, conversations and open-ended questions. The combination of in-depth qualitative analysis and some statistical data enabled the researchers to focus on underlying meanings behind responses.

#### 2.5.1. Samples

A total of 317 people completed the survey, 212 from the Southwest city of Southampton, Hampshire, and the remainder from the proximal surrounding area (UK). Southampton has a population of nearly 250,000. Nearly 90% of residents are white (compared to 82% UK), just under 46% are Christian and 46% married (both comparable to the UK) ([48]). The age range of respondents was 16–83 years. Nearly 75% were British, and 10% identified as an ethnic minority with a wide range of nationalities represented. Over 77% identified as female; nearly 70% of those identified as heterosexual and 20% as bisexual. Fifty four percent of the sample considered themselves a disabled person; of these 170 individuals, the most cited disability was ‘mental illness/nervous disorder’ (87%), with mobility impairment (28%) and Autism (27%) also common. As such the respondents were representative of our target population and national demographics, with the exception of sexual orientation where our figures differed from the national average (89% heterosexual). The findings indicated some people may not identify with certain groups (e.g., a disabled person may not feel they belong to the disabled community), and intersectionality was acknowledged in the data, i.e., some people belonged to more than one group (e.g., BME and disabled). These data are summarized in Table 1.

In addition to the survey, two professionals attended a focus group and two victims/survivors were interviewed. Both professionals worked in the same local authority but in different roles, one (“Fay”) as an Independent Domestic Violence Adviser providing emotional and practical support for victims/survivors and the other (“Louise”) worked in housing. Of the victims/survivors, one identified as BME (non-disabled) and one as BME and disabled (“Maya” and “Rhea”, respectively). Both identified as heterosexual.

#### 2.5.2. Quantitative Analysis

Data cleansing ensured accuracy, and subsequent descriptive and inferential statistics were undertaken to identify patterns and relationships indicating support needs and barriers (further information is available upon request). Due to the small numbers across each group and the intersectionality of participants, there are not enough data to identify any statistically significant patterns.

#### 2.5.3. Qualitative Analysis

Focus group and interview data were transcribed, and together with the qualitative survey data were coded and analyzed using Reflexive Thematic Analysis ([10]). For the qualitative responses to the survey data, the research team compiled a codebook (or coding ‘template’) as a tool to assist in the coding, drawing on existing knowledge and informed by coding of the interviews and focus groups ([10]; [33]). This enabled a more structured approach to coding and theme development and provided insight and symmetry through the coding process. Codes were then grouped around a central topic into overall themes. For example, the code ‘having a voice’ was merged with other codes (such as ‘lack of knowledge’) into an overall theme of ‘support needs’. Theoretically the research used a constructionism framework (meaning making from the data) while acknowledging the intersectional nature of the data. As such it recognized findings from previous studies and existing theory; however, it acknowledged these are not exhaustive so therefore additionally utilized perspectives from the participants themselves. Drawing on pragmatism and phenomenology ensured that methods aligned with research questions, lived experiences were prioritized and findings could be applied in practice.

## 3. Results

Section 2.5.1 clarifies participant data for each method independently. Table 1 gives a summary of key demographics. From here on, overall themes from all data are discussed holistically. Unless specified, findings are from the survey; results from the focus group or interviews are referred to explicitly where used. In relation to support accessed (Table 2), 70% of the respondents had accessed some support from DA services; for 63% this was in relation to their own needs. Of those, 64% of the LGBTQIA+, 12% of the BME and 57% of the disabled people had used DA services, suggesting a potential lack of engagement or desire for support services from the BME group. Over half of the participants accessed counselling services: 40% mental health support and 34% had sought support regarding how to leave their partner or report an offence.

### 3.1. Identified Themes

#### 3.1.1. Good Practice

Of those who had utilized support, over two-thirds stated it was either partially or very useful, particularly in relation to emotional and non-judgmental support, including the following example:

“*Without their support I would not have been able to leave, and I would likely not be alive*”(Fleur, European)

Findings from the focus group recognized the need for extended areas of support from family, communities and religious groups. They highlighted the importance of increased efforts to work collaboratively with other agencies, to address perpetrator behavior (through criminal justice and intervention programs) and support service users in their own homes:

“*I think one of the shifts … to attempt as much as possible to keep people in their own home, that managing a move is not necessarily the best option*”(Fay)

Interview findings also highlighted good practice by certain individuals in the police (e.g., giving advice), social work (e.g., taking allegations seriously), at court (e.g., a judge taking children into chambers) and by support agencies (e.g., them listening, understanding, providing courses). Victim/survivor interviews also highlighted the importance of lived experience support, such as having a support worker with a similar religious background:

“*I don’t have to mention to her, she already knows things… she also gave me a number for one of the Imam… even helped me with what sort of questions that I should be asking… That was that was really good*”(Maya)

#### 3.1.2. Secondary Victimization

As well as the positive experiences outlined above, some respondents described being re-victimized by agencies, services and individuals they initially anticipated would support them, leaving them feeling let down:

“*I was told by NHS officially: The abuse is too vile for them to handle*”(Maria, black, British, heterosexual, non-disabled)

“*I was not listened to… I was asked in front of my abuser every time to talk about what was happening*”(Francis, white, British, heterosexual, non-disabled)

Here it is clear that not only is there revictimization but also a lack of empathy and understanding of the power the professionals hold and the impact of their words.

Both interviewees described similar poor or unexpected responses from agencies, sometimes making the situation worse, which also evidenced a lack of understanding of domestic abuse:

“*They [police] said it’s a civil matter. But there’s violence involved… they could have nipped in the bud they could have stopped all this*”(Rhea)

A lack of community knowledge was also mentioned and even refuges made things difficult, enhancing barriers to recovery:

“*I was bleeding… got no help… they said… you have to walk to the doctor which is so far away. I had no pushchair… I wasn’t allowed to call a taxi… can somebody get my car?... No, you’re not allowed to… they’re not making this any easier*”(Rhea)

#### 3.1.3. Barriers to Obtaining Support

Just over half (n = 160) of respondents identified barriers that reduced their access to support. The most frequent included embarrassment or shame (68%), not recognizing their abuse (67%) and fear of what might happen (66%) or of not being believed (63%) as outlined in Table 3.

Findings were largely similar for specific communities and many were also common to all groups who experience DA; however, people who identified as BME scored higher than average on fear of what might happen, denial and hoping things would change. It was also clear that there were palpable emotional barriers of self-blame, such as this quote: “*I thought it was my fault*” (Nancy, mixed, British, heterosexual, non-disabled), and shame: “*I might be making a big deal and bringing shame*” (Faiza, British, Asian, heterosexual, non-disabled) around their experiences. Reports were often related to lack of recognition or denial that they were being abused:

“*I did not recognise myself as a victim as I was not cowering in corners or being hit (well not regularly)*”(Amelia, white, British, heterosexual, non-disabled)

Amelia’s example illustrates how victims/survivors share an image of the ideal DA victim, as she did not recognize her own abuse, underlining the impact stereotypes can have on social constructs of DA and reaching out for support.

For disabled people and the LGBTQIA+ community, the highest score was fear of not being believed, as exemplified by Ella:

“*My abuser was considered everybody’s friend and a great dad. I was presented as the ’difficult’ one*” (Ella, white, British, bisexual, disabled)

This was reinforced by the actions of professionals in Miriam’s case:

“*Scared I wouldn’t be believed as the police dropped my case very quickly*” (Miriam, white, British, bisexual, disabled)

Views of others or constraints faced by professionals made experiences worse, leading to self-blame that was enhanced by the manipulation of some perpetrators and manifesting itself in ‘gaslighting’ as experienced here:

“*For many years my ex was able to use my diagnoses against me with [the] authorities… everything was justifiable on the grounds of concern for my mental health or parenting*”(Freya, white, British, heterosexual, disabled)

This layering of experiences, from lack of recognition or self-belief (compounded by gaslighting), fear of reprisals or not being believed, has a significant impact on victims/survivors’ ability to seek help.

Other barriers, which are often found amongst victim/survivor reports included worry about having their children removed, which was often further exacerbated by lack of finances:

“*The fear of losing the children was a big problem and stopped me from telling social services the truth*”(Sophia, white, British, heterosexual, disabled)

“*I couldn’t afford to leave my ex…. Where would I go? The children were being manipulated … and I wasn’t prepared to leave them where this misinformation could continue*”(Elsie, white, British, heterosexual, non-disabled)

Others discussed being isolated, with practical considerations exacerbated by cultural expectations/difficulties, such as not speaking English or maintaining social standing:

“*I hesitated to admit or seek support as I’m an Asian, so I had a lot to lose in society and community*”(Amala, British, Bangladeshi, heterosexual, non-disabled)

There were two unique barriers reported by disabled respondents in terms of accessibility and being cared for by their abusers:

“*It was hard to get there, being a wheelchair user*” (Evie, white, British, heterosexual, disabled)

“*With physical disabilities… I relied on him for support, physically, mentally and financially*” (Rosie, white, British, heterosexual, disabled)

These individual barriers were compounded for some by organizational barriers; for example, many highlighted not knowing what DA includes or what support was available.

Although some sourced support through media or agencies, many desired increased awareness and promotions in educational and healthcare settings, online, via media, on posters and in different languages. Furthermore, a repeated message from both the interviews and surveys was respondents feeling they were directed from one service to another, which led to secondary victimization through repeatedly re-telling their stories (again, this consistent with all those who experience abuse) and could have potentially trapped them in their relationship:

“*I don’t want to see 10 different people*” (Rhea)

“*I sometimes wonder if it would have been easier to stay*” (Claudia, white, British, heterosexual, non-disabled)

Like the professional who noted the importance of collaborative working, survey respondents saw the need for a coordinated approach with a central hub:

“*Too many satellite agencies and resources… how is a victim meant to know… which is the best one to turn to? Needs a nationally recognised and well publicised umbrella ‘face’ so that any person… knows exactly where to turn… an online triage process… who can help and a handholding online ‘advocate’*”(Phoebe, white, English, heterosexual, non-disabled)

Additionally related to access, numerous respondents highlighted long wait times (between 2–3 years), stating they needed services sooner as delays had severe consequences such as the following:

“*thinking of taking the abuser back*” (Nihal, Sri Lankan, heterosexual, non-disabled) or ending their life

Many respondents also wanted sessions more frequently: “*At least once a week*” (Isabel, white, British, heterosexual, disabled), outside working hours and to utilize them for longer:

“*12 or 14 or 16 [sessions] most definitely are not… enough*”(Enid, white, British, heterosexual, disabled)

A lack of follow-up support was also evidenced, with some describing not only a ‘drop off’ at the end of support, but also the emotional impact and no signposting:

“*After a few months, it suddenly disappears and you are left alone in the ruins of your life*”(Evelyn, white, British, bisexual, disabled)

“*I was not given any instructions on what to do next*”(Helvi, African, lesbian, non-disabled)

There was also a recognition that support should enable victims/survivors to be able to move on; for example, one respondent suggested introducing a

“*care plan after… to prevent them returning to their abuser*”(Charlotte, white, British, bisexual, disabled)

Worryingly, self-sacrifice was evident as participants discussed not wanting to be a burden on services, and victims/survivors also noted that a lack of resources impacted service provision. They were conscious there is not enough supply to meet demand, which can put individuals at risk if they do not access help when required.

This understanding matched the fact that the barriers highlighted in the focus group included a high demand for services, necessitating prioritization of only those at highest risk; further frustration by a lack of housing stock, particularly for transgender and/or disabled people; and a lack of ‘by and for’ services (provision provided by those with lived experiences, either in relation to DA or in terms of identity characteristics). One participant described the national picture of public services funding as follows:

“*A leaky roof’.. we are putting a bucket underneath the hole… catching as many as we can… trying to stop the place from flooding*”(Louise)

In summary there are significant barriers hindering victims/survivors in obtaining support, which appear more acute for certain groups. Whilst personal and emotional barriers may be more difficult to address, organizational barriers, such as making services accessible at the point of need and offering support for as long as required, necessitates adequate resources and further consideration of support needs of clients, to which we now turn.

#### 3.1.4. Support Needs—General

When asked what support respondents wanted, the most frequently reported services requested were access to mental health support and one-to-one counselling (see Table 4), followed by advice regarding how to remain safe and advocacy.

In free-text answers, respondents expressed a strong desire for a wider variety of services, including more outdoor activities, peer support groups and online resources. They stated the benefits of connecting remotely with one interviewee highlighting the following:

“*Sites where you had like chats where you can actually talk to people on the chat that helps massively*”(Maya)

The need for support tailored to individual needs and advocacy was seen as important:

“*An appointed advocate for each victim… would be hugely, hugely beneficial… we are already overwhelmed by just trying to survive*” (Phoebe, white, English, heterosexual, non-disabled)

“*There needs to be an organised step by step system of provision… mapped out and available for the person to move through at their own pace*” (Sophia, white, British, heterosexual, disabled)

Participants also noted the importance for professionals to take a trauma-informed approach, including tailoring their language and creating a safe space:

“*Don’t use long words, or words shortened like DA … people in a distressed state struggle to process things anyway without jargon*”(Iris, white, British, straight, non-disabled)

“*They offered to find ways for me to be more comfortable and safe… drawing or listening to background music while we talked*”(Isabel, white, British, heterosexual, disabled)

There were other important factors for accessing support reflected in the survey and interview data, such as the need for confidentiality, empathy and to feel supported and be heard:

“*The support workers… were the only ones who listened to me, believed me*” (Maria, black, British, heterosexual, non-disabled)

“*Why wasn’t there a [police] officer that had more understanding… had no idea and very, like… what’s happened tell us? There was no empathy… nothing to say… you’re protected… he cannot come back… you’re not alone… we’re here to help*”(Rhea)

There was also a desire for professionals to acknowledge the importance of meeting victims/survivors’ basic needs, for them and their children, which was particularly pertinent in relation to accommodation; one interviewee was offered a refuge place, but had to leave her male teenage child behind to access it:

“*My solicitor says to me, the best thing to do is look for a flat… how am I going to afford this? … they found this really horrible place… My kids will never come… no TV… so cold… my dad bought food… some duvets and blow-up mattresses*”(Rhea)

In summary, there are patterns in general needs of victims/survivors; for example, the challenges when there are children involved, but also individual differences and preferences for types help and forms of support. Such individualized needs can be even more acute when bespoke needs of specific communities are considered.

#### 3.1.5. Support Needs—Bespoke

##### BME

Eighty percent of the survey respondents who identified as BME believed they have specific needs for support. One BME woman described how being moved to a refuge in a predominantly ‘White’ area left her feeling ‘out of place’. Linked to barriers outlined above, respondents identified different cultural expectations can mean individuals might be less likely to seek help, and proactive suggestions included the following:

“*Perhaps being given the opportunity to speak to someone from my own cultural background who could help support me as I lost every member of my family in the process of fleeing and have been culturally ostracised by my community*”(Jasminder, British, Indian, disabled)

##### Disabled People

Disabled respondents (58%) felt disabled people have specific support needs from DA services. Comments included ensuring appropriate communication aids, producing easy read versions of resources, and ensuring accessible and suitable accommodation. There was a need for professionals to have a greater understanding of the impact of mental illness as this could impact support received:

“*Mental illness can make it… look like the victim is the perpetrator when police and people are underinformed about what mental illness and trauma look like*”(Grace, white, English, ‘other’, disabled)

Significant numbers of respondents reported increased difficulty accessing services due to a variety of mental and/or physical health conditions, feeling that they could not meet their needs:

“*The main thing I think would help is autism friendly shelter… quiet, self-contained, private places people can stay… Autistic people can’t be expected to live in ordinary shelters… it’s such a strain on an already overwhelmed sensory system*”(Ella, white, British, bisexual, disabled)

##### LGBTQIA+

Individuals of all types of sexual orientation believed LGBTQIA+ individuals have specific support needs, recognizing specific challenges including discrimination, increased isolation or relationship dynamics that may make it harder to identify the abuse and for the victims/survivors to seek help. For example:

“*They would have had a different experience of abuse as it is a different kind of relationship*”(Ellie, British, French, bisexual, non-disabled)

“*I think in same sex relationships it can be quite difficult as… this might affect traditional gender roles/identity such as male on male violence*”(Freya, white, British, heterosexual, disabled)

Again, a need for services provided by those within communities or at least with a good knowledge of experiences communities may encounter, was highlighted repeatedly:

“*She [service provider] was also part of the LGBTQIA community and I think her lived experience allowed a deeper level of understanding*”(Angela, white, British, lesbian, disabled)

This appears to reduce the need for the victim/survivor to have to explain everything, reducing their distress and enhancing their feeling of being understood and supported.

These themes are summarized in Figure 1.

The findings identified several consistent themes. Despite years of research, knowledge and practice, there are still many difficulties in victims/survivors accessing help for DA and IPVA. These can be individual, for example, from personal embarrassment or shame, or resulting from more organizational issues such as limited resources and restricted access. Yet, there are patterns in the needs identified by victims/survivors, with many highlighting similarities in what they require. Additionally, some communities appear to have bespoke needs, such as ensuring appropriate access for disabled people. Whilst some good practice by various support agencies has been identified, it is alarming to note that many challenges remain. There are significant barriers to individuals obtaining help, and secondary victimization is evident, despite improvements to training and awareness.

## 4. Discussion

Interpreting the results in relation to the original research questions posed, firstly, in terms of what people think is good about current DA support services, the findings demonstrate how service providers are offering many of the services victims/survivors need, they are held in high regard, and are providing tailored services for some LBGTQIA+, BME and disabled people. Services are deemed not only useful, but for many essential for survival, wellbeing and recovery. As such, DA support services need to continue. Secondly, in consideration of research question two, how do people think DA support services for victims/survivors could be improved, many expressed a desire for recognition of their abuse and that gaining access to services was difficult due to delays or being passed around different services. Additionally, there is evident demand for increased knowledge and promotion of what constitutes DA, what services are currently available in the community, and the provision of a centralized triage to direct victims/survivors, providing standardization and clarity to victim/survivor experiences. Resourcing, in particular lengthy waiting lists, limited access to appropriate housing and basic considerations such as access to interpreters, are significant barriers to gaining support. Ideally, optionality of service provision, such as out of hours or online support, with availability for longer periods, alongside an ongoing care plan, would be available for all at the point of need.

Poor practice by a variety of agencies was noted, some with significant negative impacts resulting in secondary victimization, as victims/survivors described being let down, necessitating the need for even greater support. This reflects previous findings by [9] ([9]) that highlighted how service responses to victims/survivors could reinforce the abuse if they were not performed with care. The need for enhanced knowledge, training and basic empathy when dealing with victims/survivors, recognizing this is their lives, is necessary for all providers who come into contact with DA victims/survivors. Multidisciplinary training initiatives should incorporate awareness and understanding of competing challenges in order to co-create pragmatic and workable solutions. In terms of future resourcing, enhancing timely access to services and promoting awareness to the general public of what constitutes DA/IPVA, is also a significant need. These difficulties are reflected in all communities rather than being pertinent in specific groups.

Additionally, in relation to research question three, do LBGTQIA+, BME or disabled people have specific challenges and support needs, overwhelmingly, most participants recognized the bespoke needs of these specific communities. Provision of, by and for services from those of similar backgrounds was repeatedly encouraged due to the greater level of knowledge and understanding they possess, and as such, it is strongly recommended in order to offer fully inclusive, empathic services that reduce, rather than add to, the trauma and difficulties encountered. Further research into other groups with bespoke needs—for example men—to ascertain what and why bespoke services may be required and the benefits of these would be worthwhile.

What is clear from the findings of this study is that some barriers identified almost 20 years ago by [20] ([20]) continue today: for example, knowledge of services, need for confidentiality, shame, and fear of being judged. The barriers found in the BME group echo those found in previous studies around cultural barriers and cultural competence ([29]); although, interestingly, the issue of institutional racism was not raised in the present findings. However, in line with [29]’s ([29]) study, the barriers were weaponized by perpetrators; for example, perpetrators ‘selling the idea’ that the woman was experiencing mental health issues, which further isolated the victims/survivors. Our findings also echo those of [46] ([46]), who found similarly that their sample did not seek help because they did not want to contribute to the negative perceptions of the LGBTQIA+ community.

Participants had very clear ideas about the type of is support needed; it should be long term with ongoing follow-up contact to deal with the effects of long-term trauma. This aligns with organizations and practitioners taking a trauma-informed approach to their work when working with victims/survivors of IPVA ([19]), including long-term support after a program ends ([52]). Another key element of the trauma-informed approach is the need for ‘cultural competence’ ([52]) and an understanding of bespoke needs, which also fits with the feedback from participants. BME individuals may have language difficulties or abuse may be normalized within their family/culture. Disabled people may have additional difficulties accessing services or may be reliant upon their abuser for day-to-day care, which can significantly increase the risk of reaching out for support and limit access to support. LGBTQIA+ victims/survivors may suffer from stereotypes regarding such relationships or have more limited support networks. Moreover, our evidence suggests victims/survivors are likely to experience poor mental health, and such an approach would also recognize that as a result of living with a person who is systematically physically and/or emotionally abusive that lines between cause and effect may become blurred.

### 4.1. Limitations of the Study

This was a small-scale, localized study in the Southwest of England that focused on identifying experiences and bespoke needs of minoritized and marginalized groups. Therefore, the findings are not generalizable and do not claim to be representative of other groups’ experiences or different geographical locations; the lived experiences of the participants were interpreted through a phenomenological and intersectional lens and attempts at statistical significance were not prioritized. Furthermore, the research was conducted within the definition of domestic abuse, which includes but is not limited to IPVA; resultingly, findings must be interpreted with caution as some respondents may be relating their experiences to perpetrators within familial settings (e.g., parent, children) rather than specifically partners or ex-partners.

### 4.2. Concluding Remarks

Our study has highlighted the importance of listening to the voice of the service users, recognizing how people with lived experience are the ‘experts’ and therefore their suggestions for services should be sought and listened to. We advocate further research in this regard; exploring solutions ‘with’, rather than researching ‘to’ or ‘for’ them. It is also important to recognize the impact is broader than upon the victim/survivor; children, family and wider friendship groups can all be caught in the web of DA, and future research considering their experiences would be worthwhile.

With so much evidence clearly identifying barriers for support and also relatively straight forward solutions, we are left with the following question: why is nothing really changing? This is a particularly pertinent question in the UK as coercive control was recognized as a crime in 2015 and the recognition of the harm to children is stated in the [14] ([14]). An obvious answer here is finite resources. However, given the harm that domestic abuse does, not just to the victim/survivor but also their children and potentially the wider family and community, focusing limited resources into victim services is short sighted. Enhancements to funding perpetrator programs can help change behavior, and considering how society as a whole contributes with socially dominant narratives around gender and ‘minority groups’ (which may in itself be a misnomer as the majority of people will fall into some category or another at some time) is required. Reducing the widespread nature of domestic abuse is a national priority ([27]), but victims/survivors from minority communities can lose out when budgets are stretched and finite. Minority groups are no less deserving of living abuse free lives, but are sometimes less heard. It is essential all voices are both heard and responded to.

## Figures and Tables

**Figure 1 behavsci-15-00103-f001:**
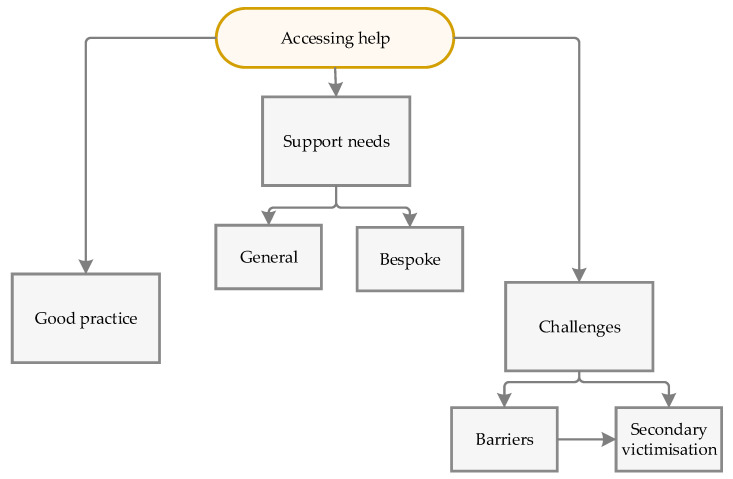
Summary of themes.

**Table 1 behavsci-15-00103-t001:** Summary of demographics.

	No. of Participants	Percentage (%)
**Simplified Ethnic Group (n = 303)**		
Not BME	241	79.5
BME	33	10.9
Unclear	29	9.6
**Gender (n = 317)**		
Female	245	77.3
Male	24	7.6
Transgender	4	1.3
Non-binary	9	2.8
Queer	3	0.9
Asexual	1	0.3
Other (nonbinary, transgender, transmasculine, and gender queer)	1	0.3
Did not answer	30	9.5
**Sexual Orientation (n = 317)**		
Straight/Heterosexual	219	69.1
Bisexual	62	19.6
Lesbian	7	2.2
Pansexual	6	1.9
Gay	5	1.6
Asexual	3	0.9
Other	3	0.9
Queer	2	0.6
Chose not to answer	10	3.2
**Disability (n = 170)**		
Mental Illness, Nervous Disorder	147	86.5
Mobility Impairment	48	28.2
Autism	47	27.6
Other Disability	24	14.1
Specific Learning Difficulty	21	12.4
Deafness (Hearing Impairment)	15	8.8
Fibromyalgia	10	5.9
ADHD	9	5.3
Blindness (Visual Impairment)	6	3.5

**Table 2 behavsci-15-00103-t002:** Summary of services accessed.

Range of Services Accessed (n = 317)		
Counselling	163	51.4
Mental Health	130	41.0
How to remain safe	113	35.6
Advocacy	110	34.7
How to leave	64	20.2
Children	61	19.2
Legal Advice	59	18.6
Reporting an offence	57	18.0
Housing	45	14.2
Physical Harm	37	11.7
Finances	17	5.4
Other	6	1.2

**Table 3 behavsci-15-00103-t003:** Barriers to accessing support.

Barriers to Accessing Support	No. of Yes Responses (n = 160)	Percentage (%)
Embarrassment or shame	109	68.1
Not recognizing abuse	107	66.8
Fear of what might happen	105	65.6
Fear of not being believed	101	63.1
Denial	90	56.3
Not a big deal	70	43.8
Hope things will change	64	40.0
Worry about information sharing	63	39.4
Love for abuser	52	32.5
Loyalty for abuser	50	31.3
Accessing support	50	31.3
Worry about losing access to children	49	30.6
Worry about finances	46	28.8
Worry about losing friends and family	46	28.8
Worry about housing	41	25.6
Other	16	10.0

**Table 4 behavsci-15-00103-t004:** Desired support.

Required Services	No. of Participants (n = 317)	Percentage %
Medical Support: Mental Health	217	68.5
Counselling	214	67.5
How to remain safe	211	66.6
Advocacy	201	63.4
How to leave a partner/abuser	197	62.1
Reporting an offence	192	60.6
Legal advice	191	60.3
Housing/access to refuge	188	59.3
Medical Support: Physical Harm	186	58.6
Finances/Money	181	57.1
Your children/dependents	180	56.8
Other	16	5.0

## Data Availability

Data is contained within the article.

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
