# Peer review of "Contemporary Treatment of Crime Victims/Survivors: Barriers Faced by Minority Groups in Accessing and Utilizing Domestic Abuse Servicesâ€"

_behavsci, 2025, doi:10.3390/bs15020103_

Round 1
Reviewer 1 Report
Comments and Suggestions for Authors
We commend the authors for addressing such an important topic and for approaching it with a focus on social impact and improving people’s lives. Their work helps identify key practices that can be applied to enhance future interventions and drive meaningful change.
Below, we outline some areas for improvement in the article:
1. Justification of Study Focus
- It is essential to better justify why these specific groups, which are highly diverse, were chosen for the study. While the author identifies prevalence percentages of violence within these groups using various surveys, these percentages are not compared with more mainstream groups. This raises the possibility that the identified prevalence could be attributed to their vulnerability as opposed to direct experiences of violence. The intersectionality of these groups makes them important to study, but further justification for their selection is needed.
2. Broader Scope of Violence
- Consider including information highlighting that violence against women is not limited to cohabiting partners. In fact, a significant amount of gender-based or sexual violence occurs in dating contexts or casual relationships.
3. Relevance of Barriers to Support
- Some barriers mentioned, such as the fear of losing custody of children, are shared with other groups. The relevance of these barriers to the studied groups specifically should be better justified to strengthen the argument.
4. Theoretical Framework
- The theoretical framework should clarify that the arguments presented are based on prior studies or theories but are not exhaustive. This distinction would help readers understand the context of the claims made.
5. Research Questions
- The research questions lack scientific orientation. For example, terms like "doing well" are subjective and not suitable for scientific inquiry. Instead, the questions could focus on identifying factors that foster inclusion within these groups or help overcome specific challenges, as well as those that hinder such progress.
6. Positive Acknowledgment of Practical Orientation
- The article’s orientation toward analyzing activities and tools that promote inclusion and social impact is commendable. The authors are to be congratulated for going beyond mere analysis to propose actionable solutions to improve these situations scientifically.
7. Methodology
- The questionnaire section would benefit from more detail to enhance replicability. For instance, specify the questions asked and the type of data sought.
- Similarly, the focus group methodology needs to be expanded. Currently, it only mentions difficulties in engaging organizations but does not elaborate on the reasons or the types of focus groups conducted.
- Including a summary table of participants across methodologies (e.g., sociodemographic variables for survey respondents, and profiles of focus group and interview participants) would add clarity.
- It would also be helpful to provide the coding framework or key questions used in interviews to improve transparency and replicability.
- Some sociodemographic information currently presented in the results section belongs in the methodology, as it describes the sample.
8. Regional Context
- The region where the study was conducted should be explicitly stated to provide contextual relevance.
9. Organization of Results
- The integration of focus group and interview discourse with survey data makes the analysis challenging to follow. At times, themes are blended, and the depth of analysis is compromised. Despite this, the quotations are valuable and rich in information, identifying practices that could improve lives. However, there is an over-reliance on quotations with insufficient accompanying analysis, leaving the reader to draw conclusions independently.
- The results would benefit from clearer organization, deeper analysis, and less reliance on raw quotations.
10. Writing Style
- The writing style would improve with a more natural and coherent flow. Currently, the article reads like a list of bullet points with abrupt topic transitions. Guiding the reader more effectively through the themes would enhance the overall readability.
11. Conclusions and Discussion
- The conclusions and discussion sections are scattered, lacking focus on the main objectives of the article. Reiterating the research questions and linking them to the findings would provide more cohesion. This would help clarify the key messages and contributions of the article.
12. Proofreading
- A thorough proofreading is necessary to address errors such as “Mminority groups” on line 557.
Overall Recommendation The article demonstrates a solid fieldwork foundation and addresses an important topic, making it publishable with revisions. However, the writing needs refinement to achieve the scientific rigor required by the journal. Enhancing the methodology, improving the depth of analysis, and refining the presentation of results and writing style would significantly elevate the article.
Reviewer 2 Report
Comments and Suggestions for Authors
Thank you for offering me the opportunity to review this article. As an ex practitioner in the field I concur that it is a very important and useful area that requires much additional exploration. However, in its current form I believe this article falls short in delivering its objectives and needs major revisions before it will be ready for publication. I understand the authors may choose not to conduct the additional analysis but without it I cannot see that this article provides the necessary rigor to add significant value in its current form to such an important issue. To publish it without this will not do it justice in my humble view.
My reasons for this are listed as follows:
1. The most significant flaw at this stage relates to the data sources.
a. Issue 1 is that the focus group and interviews are too few to draw any meaningful conclusions. In total, it is just 4 participants. As such, almost anything they say cannot be viewed as representative. I would suggest their inclusion is reconsidered as a large bulk of the article draws upon this information to inform subsequent overarching discussion points and with such a small number of participants it undermines the validity of the discussion and any conclusions or recommendations.
b. The quantitative data is much more in terms of volume, but is barely examined in any detail. The analysis method is skirted over in a couple of sentences (what is presented is purely descriptive). What methods/stats tests were used and why? Are the results significant or not? Considering the bulk of the data comes from this section it is grossly disproportionate to the paragraph afforded to describing the thematic analysis of the interviews.
i. Additionally, how do the findings differentiate between the examined groups. This is key. The article needs to explore if there are distinctions in responses between BME/disabled and LGBTQIA+ groups. This is where I suggest the real value in this study will lie. To do this, the authors will have to conduct much more data analysis.
2. In respect of other less significant issues, the distinctions between DA and IPV are not clear throughout. The opening introduction explores it a little, but does not clarify that DA can be interfamilial and that IPV is specifically partner related. This needs clarifying much better.
3. The subsequent sections in the introduction that relate to the different groups i.e. BME/disabled and LGBTQIA+ also fails to differentiate between DA and IPV. I acknowledge that the articles focus is on the broader issue of DA, but many of the stats appear to relate to IPV, especially those in the LGBTQIA+ section. The distinctions in this regard need to be much clearly stated to improve the transparency of this section. As a minimum it needs highlighting as a significant limitation if this cannot be achieved as it may misrepresent the issues of victims and survivors from each group.
4. The barriers section could be amalgamated into a single subsection as some have just 1-2 sentences.
5. How does the geographic area the study was completed affect the representative nature of the findings? This is only alluded to but not specified. I would expect exact details, most likely from census data to support this if it is available. It certainly will be for the BME and disabled groups.
6. Based on all of the aforementioned, the discussion sections validity is significantly undermined and this should be completely rewritten if the authors choose to complete the additional analysis required to make the article more relevant.
